# Smooth 3D Path Planning by Means of Multiobjective Optimization for Fixed-Wing UAVs

**Franklin Samaniego *** , **Javier Sanchis** , **Sergio Garcia-Nieto** and **Raul Simarro**

Instituto Universitario de Automática e Informática Industrial, Universitat Politècnica de València, 46022 Valencia, Spain; jsanchis@isa.upv.es (J.S.); sgnieto@isa.upv.es (S.G.N.); rausifer@isa.upv.es (R.S.)

\* Correspondence: frasarie@doctor.upv.es; Tel.: +593-99-130-5550

**Abstract:** Demand for 3D planning and guidance algorithms is increasing due, in part, to the increase in unmanned vehicle-based applications. Traditionally, two-dimensional (2D) trajectory planning algorithms address the problem by using the approach of maintaining a constant altitude. Addressing the problem of path planning in a three-dimensional (3D) space implies more complex scenarios where maintaining altitude is not a valid approach. The work presented here implements an architecture for the generation of 3D flight paths for fixed-wing unmanned aerial vehicles (UAVs). The aim is to determine the feasible flight path by minimizing the turning effort, starting from a set of control points in 3D space, including the initial and final point. The trajectory generated takes into account the rotation and elevation constraints of the UAV. From the defined control points and the movement constraints of the UAV, a path is generated that combines the union of the control points by means of a set of rectilinear segments and spherical curves. However, this design methodology means that the problem does not have a single solution; in other words, there are infinite solutions for the generation of the final path. For this reason, a multiobjective optimization problem (MOP) is proposed with the aim of independently maximizing each of the turning radii of the path. Finally, to produce a complete results visualization of the MOP and the final 3D trajectory, the architecture was implemented in a simulation with Matlab/Simulink/flightGear.

**Keywords:** UAV; path planning; smooth path planning; multiobjective optimization

## 1. Introduction

The latest technological and scientific advances in the field of mobile robotics have enabled the area of autonomous vehicles (AV) to become a reality applied to the civil and military sectors [1,2]. Consequently, this technological branch presents a constant and vertiginous development in different fields, among them the development of new navigation and guidance techniques. The constant evolution in this field responds to new challenges in real applications [3–5].

Autonomous vehicles with non-holonomic characteristics [6–8] can be technologically adapted to different environments, and so, produce unmanned ground vehicles (UGVs) [9–11], unmanned underwater vehicles (UUVs) [12–14], and unmanned aerial vehicles (UAVs) [15–18]. Obviously, each of the above categories presents its own scientific and technological challenges, including path planning, navigation, and guidance.

It is important to note that the most common problem when determining a possible feasible path is the consideration of the AV intrinsic constraints. Therefore, the non-inclusion of kinematic and/or dynamic constraints of the AV when addressing the path planning problem may lead to non-feasible solutions that, for example, make it impossible for the AV to satisfactorily follow a path. However, including in the design all the constraints of the AV in the calculation phase of the path

planning can lead to very complex optimization problems without a single solution and with very high computational costs.

This paper focuses on the generation of smooth paths for fixed wing unmanned aerial vehicles (UAVs) that have high flight capabilities and extended flight-time missions, even in low power propulsion situations [19]. For these reasons, fixed-wing UAVs are suitable for use in terrain mapping applications for later action by the security forces, and in search and rescue tasks, both for the detection of people and the provision of first aid.

Due to the non-holonomic constraints of fixed-wing UAVs, the aim is to create a smooth three-dimensional curve from an initial point to a goal point through a complex 3D space with or without obstacles. To achieve this goal, it is essential to define a feasible path that minimizes flight turning effort and distance traveled. The ordered set of waypoints that will be used to generate the path to follow is defined as the control points set. In general, the problem of path planning is defined in a space region denoted as $\chi$ and split as the tripla $(\chi_{free}, \chi_{init}, \chi_{goal})$. The movement space is defined as $(\mathbb{W} = \mathbb{R}^N : N = \{2, 3\}$, where the obstacle region is denoted by $\chi_{obs}$, so that $\chi / \chi_{obs}$ is an open set denoting the collision-free space $\chi_{free}$. The initial condition $\chi_{init}$ and the final condition $\chi_{goal}$ are elements of $\chi_{free}$.

The set of control points that define the collision-free space is calculated using specific path planning methods based on continuous and discrete environment sampling. Some examples of these techniques are: the rapidly-exploring random tree (RRT) [20–23]; probabilistic road maps (PRM) [24–28]; heuristic planners (genetic algorithms—GA) [29,30]; swarm intelligence [31–34]; fuzzy logic [35,36]); Voronoi diagrams [37–39]; artificial potential [40–43]; and recursive rewarding modified adaptive cell decomposition (RR-MACD) [44].

All the techniques mentioned build piece-wise paths in 2D or 3D to address the standard problem of path planning. These methods may provide optimal or near-optimal paths; however, they cannot guarantee smoothness and continuity, which could make it difficult to guide the UAV through the paths generated. Moreover, these techniques do not directly incorporate the operational constraints of the UAV and the environment. Therefore, this paper proposes a methodology to define feasible and smooth UAV paths, including system operational kinematic constraints.

The ability of an UAV to fly from one position to another and the consequent definition of the mission to be performed remains a challenge that requires the application of increasingly sophisticated strategies. One of the fundamental constraints to be considered in mission planning is the ability to be positionable and sensorially oriented (on the UAV or the environment) throughout the duration of the mission. This sensory location enables the construction of maps of the environment, and also enables the UAV to estimate its own current position and complete its self-location on the map. Similarly, it is important to note that good positioning and sensory orientation can be achieved by making a path plan and tracking it across or beyond the sensory detection domain. It is important to mention that the definition of smooth paths is not a new subject; various approaches have been proposed for non-holonomic UAVs, such as Dubins [45,46] in which paths are defined by connecting lines and arc-circular segments. The disadvantage is the generation of discontinuities in the connection points between segments. Another useful methodology in the literature is Clothoid curves [47–49], the main advantage being increases in curvature as a function of the arc-length; meanwhile the disadvantage lies in the limitation of length. These approximate curves generate continuities [50] of the type $C^1$ (a continuous path $C^1$ preserves the continuity of the tangent vector, in addition to maintaining continuous speed) and have been used in applications on mobile vehicles and on UAVs with flight limitation at a constant altitude, and examples are mentioned in [51–55]. An intuitive approach that ensures $C^2$ continuity (a continuous trajectory $C^2$ preserves the second order differential values at each point of the trajectory, in addition to maintaining the continuity of the acceleration vector) in 3D planning focuses on curvature and torsion zero at the junction points, as proposed in [56]. This is an interesting proposal where the 3D curve is built from a 2D curve in a $(x, y)$ plane; the resulting length of this curve is the main parameter to build the curve in the other $(x, z)$ plane. Finally, the Bézier,

B-spline [57,58] are easy to implement polynomial curves; however, none are suitable for planning since they are sensitive to control parameters and weights [59], and do not take into account the constraints of the vehicles on which they are applied, so those curves need to be optimized. Finally, these types of parametric curves have no physical meaning and the relationship between design parameters and system variables is not defined. The above-mentioned approximation methodologies generate two different phenomena called interpolation (generation of a curve which must pass through the control points) and approximation (generation of a curve that approximates the control points, but may only go through the start and finish rather than all of them) detailed in [60,61]. The work presented here explores both phenomena in-depth to respond to the definition of feasible trajectories.

The starting point of this work is the set of 3D collision-free points generated by the various planners that guarantee that the path departs from the init point and ends at the goal point, and avoids static and dynamic obstacles. From these collision-free points, an ordered set of straight lines is built that define a first path that will later be smoothed to incorporate the feasibility and constraints of the UAV. The UAV constraints in this work are focused on its ability to turn horizontally and vertically. Therefore, for a UAV to complete a sequence of turns at a defined speed, it must determine its minimum turning radius $Rp$. If the turning radius is too small, the UAV will lose the trajectory; however, if the turning radius grows, the UAV can perform maneuvers with less effort.

The aim is to maintain 3D planning results, and at the same time, generate a finite set of possible 3D curves that optimize an approximate 3D curve, and simultaneously, the turns of the UAV—which is raised as a multiobjective optimization problem (MOP) [62]. This approach will result in a set of paths that meet UAV constraints expressed as dominant solutions on a Pareto $n$ dimensional front [63]. Finally, selection criteria must be applied to determine the desired response from the point of view of curvature $\kappa$ and torsion $\tau$ of the generated 3D curve. Thus, in order to verify the functionality of the proposed methodology, the results of the curves generated after the 3D curve optimization were compared with a known Bézier type approximation methodology [64].

This document is structured as follows. A brief summary of MOP concepts is given in Section 2.1. In Section 2.2, a brief description of smooth curves is given. Section 3 presents the formulation of the problem. Section 4 details the complete methodology for solving the problem. Section 5 details the experiments and results of 3D smooth path planning. Finally, conclusions and future work are presented in Section 6.

## 2. Background

### 2.1. MultiObjetive Optimization

The optimization problem (OP) attempts to determine a solution that represents the optimal value (minimum or maximum) of a function, such as $f : X \to \mathbb{R}$, where $X$ is a feasible decision vector, being $\min(f(x)) : x \in X$. However, for problems where simultaneous optimization of more than one objective is necessary, i.e., multiobjective optimization (MOP), the function is shaped $f : x \to \mathbb{R}^k$, where $k \geq 2$ is the number of objectives. Therefore, the value vector of the target function could be defined as $f : X \to \mathbb{R}^k, f(x) = (f_1(x), \cdots, f_k(x))^T$. However, there is not usually a single $X$ that generates an optimum that simultaneously satisfies each of the $k$ objectives, due to the conflict between the objectives. The aim is to find a situation in which all objectives are satisfactorily within acceptable parameters. The MOP solution leads to points where any improvement in one target results in the degradation of any other target (one or more). Thus, these points are represented as a Pareto front [63], where all the points of the front are equally optimal.

Therefore, as expressed in [62] the MOP can be established as

$$\min \mathbf{J}(\theta) = \min_{\theta \in D}[J_1(\theta), J_2(\theta), \cdots, J_m(\theta)] \tag{1}$$

subject to:

$$g(\theta) \leq 0$$
$$h(\theta) = 0 \tag{2}$$
$$\theta_{il} \leq \theta_i \leq \theta_{iu}, i = [1, \cdots, n],$$

where $\theta \in \mathbb{R}^n$ is the decision vector, $D$ is the decision space; $\mathbf{J}(\theta) \in \mathbb{R}^m$ is the target vector; $g(\theta)$ and $h(\theta)$ are constraint vectors; and finally, $\theta_{il}$ is the upper boundary and $\theta_{iu}$ is the lower boundary of the decision space. Consequently, there is no single optimal model; in fact, there is a set of optimal solutions with different trade-offs between objectives, where none is better than the others. Using the definition of dominance, the Pareto set $\Theta_P$ is the set of each non-dominated solution.

In this way, the Pareto domination is defined in case a solution $\theta^1$ dominates another solution $\theta^2$; that is, $(\theta^1 \prec \theta^2)$, if

$$\forall i \in B, J_i(\theta^1) \leq J_i(\theta^2) \wedge \exists k \in B : J_k(\theta^1) < J_k(\theta^2), \tag{3}$$

where $J_i(\theta), i \in B := [1 \cdots m]$ are the objectives to be optimized. Therefore, the optimal set of Pareto $\Theta_P$ is given by

$$\Theta_P = \theta \in D | \nexists \tilde{\theta} \in D : \tilde{\theta} \prec \theta$$
$$J(\Theta_p) = \{J(\theta) | \theta \in \Theta_p\}, \tag{4}$$

where $\Theta_p$ and $J(\Theta_p)$ are MOP solutions. However, in most cases they are unreachable because $\Theta_P$ normally includes infinite solutions. Therefore, a finite set of $\Theta_P^*$ from $\Theta_P$ and another finite set of $J(\Theta_p^*)$ from $J(\Theta_p)$ represent satisfactory solutions. Starting from $J(\Theta_p^*)$, the decision maker selects a solution according to the established preferences. For example, a certain point in the Pareto front that is close to the ideal point (called utopia point) $\mathbf{J}^{ideal}$.

$$\mathbf{J}^{ideal} = \{J_{1\ min}(\theta), \cdots, J_{m\ min}(\theta)\}. \tag{5}$$

Hence, an appropriate methodology for characterizing MOP is known as the elitist multiobjective evolutionary algorithm ($\epsilon^\nearrow-MOGA$) [62], which makes a distributed approach to Pareto's front. The aim of $\epsilon^\nearrow-MOGA$ is to find an intelligent distributed convergence towards a set of $\epsilon$-Pareto; i.e., determine $\theta_{P\epsilon}^*$ along the Pareto front $\mathbf{J}(\Theta_P)$. The target space is split into a fixed number of *boxes*. Therefore, for each dimension $i \in B$, cells $n\_box_i$ wide are created $\epsilon_i$, where

$$\epsilon_i = \left(J_i^{max} - J_i^{min}\right) / n\_box_i$$
$$J_i^{max} = \max_{\theta \in \Theta_{P\epsilon}^*} J_i(\theta), \ J_i^{min} = \min_{\theta \in \Theta_{P\epsilon}^*} J_i(\theta). \tag{6}$$

Each *box* can be occupied by a single solution; therefore, this grid produces an intelligent distribution and preserves the diversity of $\mathbf{J}(\Theta_{P\epsilon}^*)$. In addition, it is important to note that only the occupied boxes are verified, avoiding the need to use other clustering techniques to obtain adequate distributions. On the other hand, for a solution $\theta \in D$, $box_i(\theta)$ is defined as

$$box_i(\theta) = \left\lceil \frac{J_i(\theta) - J_i^{min}}{J_i^{max} - J_i^{min}} \cdot n\_box_i \right\rceil \forall_i \in B \tag{7}$$

Then, $\mathbf{box}(\theta) = \{box_1(\theta), \cdots, box_m(\theta)\}$. A solution $\theta^1$ with value $\mathbf{J}(\theta^1)$ $\epsilon$ dominates the solution $\theta^2$ with value $J(\theta^2)$, denoted by $\theta^1 \prec_\epsilon \theta^2$, only if

$$\mathbf{box}(\theta^1) \prec \mathbf{box}(\theta^2) \vee \left(\mathbf{box}(\theta^1) = \mathbf{box}(\theta^2) \ and \ \theta^1 \prec \theta^2\right). \tag{8}$$

So, a set $\Theta^*_{P\epsilon} \subseteq \Theta_P$ is $\epsilon$-stop only if

$$\forall \theta^1, \theta^2 \in \Theta^*_{P\epsilon}, \theta^1 \neq \theta^2, \mathbf{box}(\theta^1) \neq \mathbf{box}(\theta^2) \; and \; \mathbf{box}(\theta^1) \nprec_\epsilon \mathbf{box}(\theta^2). \tag{9}$$

Thus, a $\Theta^*_{P\epsilon}$ is reached with the greatest possible number of solutions that adequately characterize Pareto's front and whose number of possible solutions depend on $n\_box_i$, and will not exceed the following level.

$$|\Theta^*_{P\epsilon}| \leq \frac{\Pi^n_{i=1} n\_box_i + 1}{n\_box_{max} + 1}, n\_box_{max} = \max_i n\_box_i. \tag{10}$$

Hence, it is possible to control the maximum number of solutions to characterize the Pareto front. Finally, due to the definition of the box, the anchor points $J_i(\theta^{i*})$ are assigned a value of $box_i(\theta^{i*}) = 0$, whereby $J_i(\theta^{i*}) = J^{min}_i$. Therefore, no solution $\theta$ can $\epsilon$-dominate them because, by applying the definition of the box, their $box_i(\theta) \geq 1$.

The above process delivers two defined sets of responses: (a) the Pareto front $\Theta^*_P$ deploys a finite set of minimum values as the optimal path response within the search space; (b) the corresponding optimal points $J(\Theta^*_j)$.

*2.2. 3D Curves for UAVs*

A non-holonomic UAV [65] can perform flights in 3D Euclidean space. Nevertheless, to complete each movement sequence (horizontal and vertical), a set of UAV flight constraints must overcome. An UAV attempts to perform 3D movements at a defined velocity, meaning that the UAV is moving continuously, attempting to maintain that velocity. However, an UAV has a maximum capacity of turn and elevation at a defined velocity. Therefore, the aim is to build a 3D smooth curve inside the UAV flight turning boundaries, in such a way to reach a complete 3D smooth curve tracking.

A smooth curve can be defined as the representation of a continuous function, which can be expressed as $C : I \to X$, where $I$ is the curve interval composed of real numbers, while $X$ represents the topological space. If the topological space is three-dimensional $X = \mathbb{R}^3$, then $C : [a, b] \to \mathbb{R}^3$, is a differentiable injectable and continuous function, whose arc-length $s$ is independent of the parametrization $C$. If a UAV is considered as a particle that travels along the envelope of the $C$ curve at a defined speed $v$, this particle suffers changes in its local coordinate system due to the set of rotations of the $C$ curve. Therefore, $C$ must not allow rotation changes outside of the intrinsic rotation capabilities of the UAV.

It is important to mention that the formulation known as Frenet–Serret [66,67] describes the kinematic properties of particles that move along three-dimensional Euclidean space $\mathbb{R}^3$ continuous and differentiable $C(s)$, and parametetrized by its arc-length $s$ (the arc-length is an invariant Euclidean characteristic of the curve). However, if we assume a curve given by a series of points along the Euclidean space as $r(t)$, where the parameter $t$ does not need the arc-length, then the tangent (T), normal (N) and bi-normal (B) derived vectors can be described, as all are mutually perpendicular (orthogonal base). Therefore, according to the theory of differential geometry of curves [68], the following equals can be defined as:

$$T(t) = N(t) \times B(t) = \frac{r'(t)}{\|r'(t)\|}$$

$$B(t) = T(t) \times N(t) = \frac{r'(t) \times r''(t)}{\|r'(t) \times r''(t)\|} \tag{11}$$

$$N(t) = B(t) \times T(t) = \frac{[r'(t) \times r''(t)] \times r'(t)}{\| [r'(t) \times r''(t)] \times r'(t)\|}$$

where $r'(t) = \frac{dr(t)}{dt}$, $r''(t) = \frac{d^2 r(t)}{dr^2}$ and $r'''(t) = \frac{d^3 r(t)}{dr^3}$ are the position vector derivatives $r(t)$. These three vectors configure a navigation reference system of the UAV. Similarly, it is important to mention that

the tangent vector $T(t)$ is parallel to velocity, while the normal vector $N(t)$ is represented by the change of direction per time unit of velocity.

The curvature terms $\kappa$ (change of direction of the vector tangent $T(t)$ to the curve $r(t)$) and torsion $\tau$ (change of direction of the vector bi-normal $B(t)$) are defined as:

$$\kappa = \frac{\|r'(t) \times r''(t)\|}{\|r'(t)\|^3} \tag{12}$$

$$\tau = \frac{r'(t) \cdot [r''(t) \times r'''(t)]}{\|r'(t) \times r''(t)\|^2}. \tag{13}$$

$\kappa$ indicates a direct correlation with the horizontal rotation capability of the UAV, while $\tau$ indicates the elevation capability of the UAV. Therefore, the triedro Frenet–Serret can be defined in matricial notation as a skew-symmetric matrix:

$$\begin{bmatrix} \dot{T} \\ \dot{N} \\ \dot{B} \end{bmatrix} = \begin{bmatrix} 0 & \kappa & 0 \\ -\kappa & 0 & \tau \\ 0 & -\tau & 0 \end{bmatrix}, \tag{14}$$

where the point over the variable indicates the derivative with respect to the parameter of arc-length $s$.

In summary, the space curve according to the formulation Frenet–Serret, defines a smooth curve that will be built from a spatial point and its tangent vector; this curve will be generated in the Euclidean 3D space depending on the pre-defined values of $\kappa$ and $\tau$, and finalize at a spatial 3D point after completing the arc-length $s$. However, our goal was to start from a defined point $p_{init}$, and build a curve that touches a target point $p_{goal}$; therefore, the values of $\kappa$ and $\tau$ have to fit so that when complete, the arc-length $s$ will touch $p_{goal}$. That gives us an infinite number of possible values of $\kappa$ and $\tau$ with which we could meet that goal. Even if the maximum and minimum values of $\kappa$ and $\tau$ are bounded, the complexity of the problem is high. In [69] it is mentioned that the complexity in a 2D environment becomes NP-hard, and the need is demonstrated for path planning algorithms that generate short paths with bounded curvatures in complicated environments. The aim is to find for possible approximate values of $\kappa$ and $\tau$ that do not exceed the turn capabilities of the UAV. In other words, starting from Figure 1, and assuming a configuration of the UAV as a triplet $(\rho, \kappa, \tau)$, where $\rho = [P_1, \cdots, P_5]$ is a dimensional vector that specifies $n$ collision-free points, $\kappa$ and $\tau$ are a set of curvatures and torsion along the path. The smooth curve starts from $P_1$ and reaches out to $P_5$, and approaches the remaining $\rho$ without affecting them, with values of $\kappa$ and $\tau$ within the boundaries established by the maneuverability capabilities of the UAV. In summary, the selection of $\rho_{goal}$ points which determine radii of the tangent curves to the $\rho$ points, will be obtained by solving a multiobjective optimization problem (MOP). This MOP is stated in such a way that the values of all radii will be maximized simultaneously. Obviously, the optimizer handles these values, taking into account that they are in conflict (as radius of one of them is increased, consequently, the adjacent radius are reduced. Therefore, the MOP solver will try to find a trade-off solution that guarantees the best set of points $\rho_{goal}$ for all tangent curves between control points $\rho$.

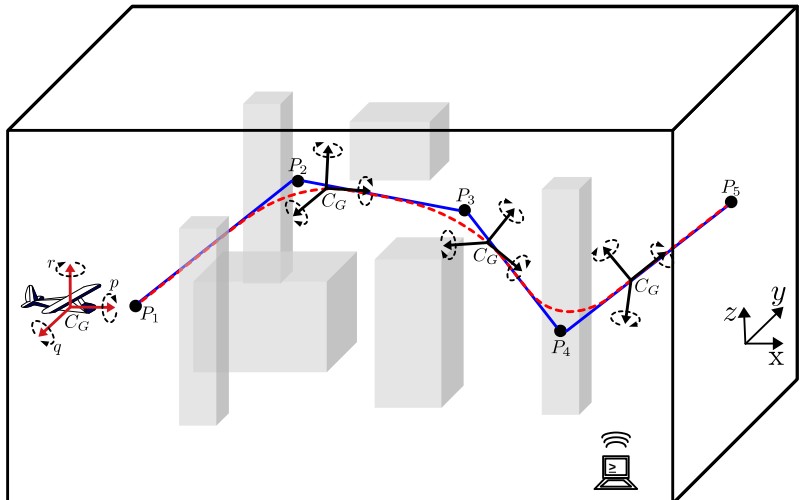

**Figure 1.** Perspective of the 3D flight problem for a fixed wing UAV, where $C_G$ represents the position vector of the center of gravity of the UAV. The global coordinate system $CS_g$ is placed at the origin, the orientation of the local body coordinate system $CS_b$ expressed by Euler roll, pitch and yaw angles respectively, which have been defined by three unitary orthogonal vectors aligned with the three axles of the vehicle and centered at $C_G$. Finally, the angular velocities along the local axis of the UAV $X$, $Y$ and $Z$ are represented by $p$, $q$ and $r$, respectively. The set of collision-free points $P_i$ is represented by black dots; the blue line describes the discrete path built from 3D path planning; the red dotted line is the new smooth optimized path that the UAV could follow.

## 3. Problem Definition

Let us assume a workspace $\mathbb{W} = \mathbb{R}^3$, where it is possible to define a set of static or dynamic obstacles, such as ground or aerial boxes of different dimensions and locations (see Figure 1). The in-flight UAV receives data from its control station regarding environmental conditions and performs the necessary calculations to determine the best smooth 3D trajectory. Relevant data include the set of ordered 3D flight waypoints that are collision-free $\rho = [P_1, \cdots, P_5]$ in the environment. The intrinsic maneuverability capabilities are determined by a $Rp$ turning radius (which determines the vertical and horizontal turning limitations) defined by their flight speed. The aim is to start from $\rho_{init}$ and reach $\rho_{goal}$ in such a way that the UAV approaches the direct trajectory marked by the ordered sequence $\rho$. Therefore, $\rho = P_i(x_i, y_i, z_i)$ where $(i = 1, \cdots, n)$ and $n$ is the total set of collision-free spaces and can be expressed as a discrete interpolation sequence $\rho = f(t_i) \rightarrow \mathbb{R}$, where $f(t_i)$ is a set of nodes in 3D space. Therefore, it is possible to establish a set of sub-intervals $(n - 1)$ between $i = 1$ and $i = n$ partitioned in $[a, b]$, defined as:

$$[a,b] = [t_1, t_2] \cup [t_2, t_3] \cup \cdots \cup [t_{n-2}, t_{n-1}] \cup [t_{n-1}, t_n]$$
$$a = t_1 \leq t_2 \leq \cdots \leq t_{n-1} \leq t_n = b. \tag{15}$$

A linear union between pairs of points then results in $\boldsymbol{L} : [a,b] \rightarrow (x, y, z)$. And can be expressed as a set of straight lines that mark a direct flight path $\boldsymbol{L}(t)$ split into $(n - 1)$ piece-wise.

$$\boldsymbol{L}(t) = \begin{cases} L_1(t) : & t \in [t_1, t_2] \\ L_2(t) : & t \in [t_2, t_3] \\ \vdots \\ L_n(t) : & t \in [t_{n-1}, t_n] \end{cases} \tag{16}$$
$$\boldsymbol{L}(t) = L_1(t) + L_2(t) + \cdots + L_n(t).$$

Therefore, $L(t)$ is a linear interpolation function for the discrete sequence $\rho = f(t_i)$. In the same way, between the $\rho$ points, there is a subset of $(n-1)$ straight lines that join the init and the goal of the trajectory along the collision-free flight space.

However, a non-holonomic UAV cannot perform every type of maneuver defined by $L(t)$. In general, it is desirable to perform maneuvers with a high turning radius. Therefore, the approach presented in this work builds a smooth trajectory from $\rho$, that attempts to avoid inappropriate maneuvers using low values of $\kappa$ and $\tau$ included within the boundaries of the UAV flight turn, while simultaneously closing in on the trajectory $L(t)$.

Let us assume, from Figure 1, that the blue line denoted as $L(t)$ is the direct path between the collision-free points of the environment, and the red dotted line is a smooth 3D spatial curve defined as $C(t)$. The construction of this 3D smooth curve $C(t)$ is done by joining a set of segments that can be of two types: spherical curves $S$ (defined from a sphere of radius $Rp$) or straight lines $L$. Thus, each $S$ segment is defined by three continuous points of $\rho$; this segment $S$ has two points of tangency, one for each pair of adjacent straight lines $L(t)$ formed by the current set of three points $\rho$. Hence, each $S$ segment may have infinite solutions, with each radius $Rp$ resulting in different tangent points on the lines $L(t)$. Therefore, for each $S$ segment, an infinite set of spheres can be defined, which will be linked through the relevant $L$ segments or another $S$ segment. Obviously, this approach to the problem suggests the existence of infinite combinations for the $S$ and $L$ segments. The way to address this issue has been through the approach of an MOP.

## 4. Methodology

This section describes the proposed methodology for the generation of 3D smooth trajectories. The proposed method is split into two parts, first detailing how the $S$ segments were obtained, and then describing the union with the $L$ segments.

### 4.1. Definition of Spherical Segment

Let us assume that from the result of a path planning, $\rho = [P_1, \cdots, P_n]$ is the set of collision-free points of the environment described in Figure 2 (red points). This set of points is defined as $P_i(x_i, y_i, z_i) :$ $i = \{1, \cdots, n\}$, where $p_{init} = P_i(x_i, y_i, z_i) : i = \{1\}$ and $p_{goal} = P_i(x_i, y_i, z_i) : i = \{n\}$.

As indicated above, Figure 2b shows an osculating sphere ($oS$) [68] defined with a minimum turning radius value $Rp$, located between the set of the first 3 $P_i$ and tangential to the straight lines $L(t)$ formed between the same set of $P_i$. Therefore, taking into account the number of collision-free points $\rho$, the set of spheres is equal to $G_i : i = \{1, \cdots, n-2\}$, as shown in Figure 2c (orthogonal view).

Figure 2b shows the first $G_i : i = 1$ located among the three first $P_i$s. Therefore, it is possible to define a plane $\pi_i : i = 1$ between the same points $P_i$, which will have an angle in relation to the location of the current set of points $P_i$, as can be seen in the Figure 3a,b. The importance of the definition of this plane is given by the fact that within it is contained the center of $G_i$ with radius $Rp$. In this way, there is a self-contained curve $S_i$ (as a series of points along the Euclidean space) on the surface of the sphere and tangent to $L(t)$ with $t_2$ and $t_3$ in plane $\pi_i$, as shown in Figure 3; hence, the $S_i$ curve segment (black line) is defined as:

$$
\begin{aligned}
S_i(t) &= [S_x, S_y, S_z] \\
S_{i_x} &= x_0 + Rp * \sin(\psi) * \cos(\varphi) \\
S_{i_y} &= y_0 + Rp * \sin(\psi) * \sin(\varphi) \\
S_{i_z} &= z_0 + Rp * \cos(\psi)
\end{aligned}
\left.\begin{aligned}\\\varphi_1 \geq \varphi \geq \varphi_2 \\ \wedge \\ \psi_1 \leq \psi \leq \psi_2.\end{aligned}\right\}
\tag{17}
$$

where, $x_0$, $y_0$ and $z_0$ together represent the center of $G_i$. The curve $S_i$ performs a horizontal and vertical path due to the angular ranges of $\psi$ and $\varphi$, which implies variations in the values of $\kappa$ and $\tau$ (these have a direct connection to $Rp$ and the arc-length of $S_i$). Consequently, if the value of $Rp$ grows, $S_i$ also grows, while $\kappa$ and $\tau$ decrease.

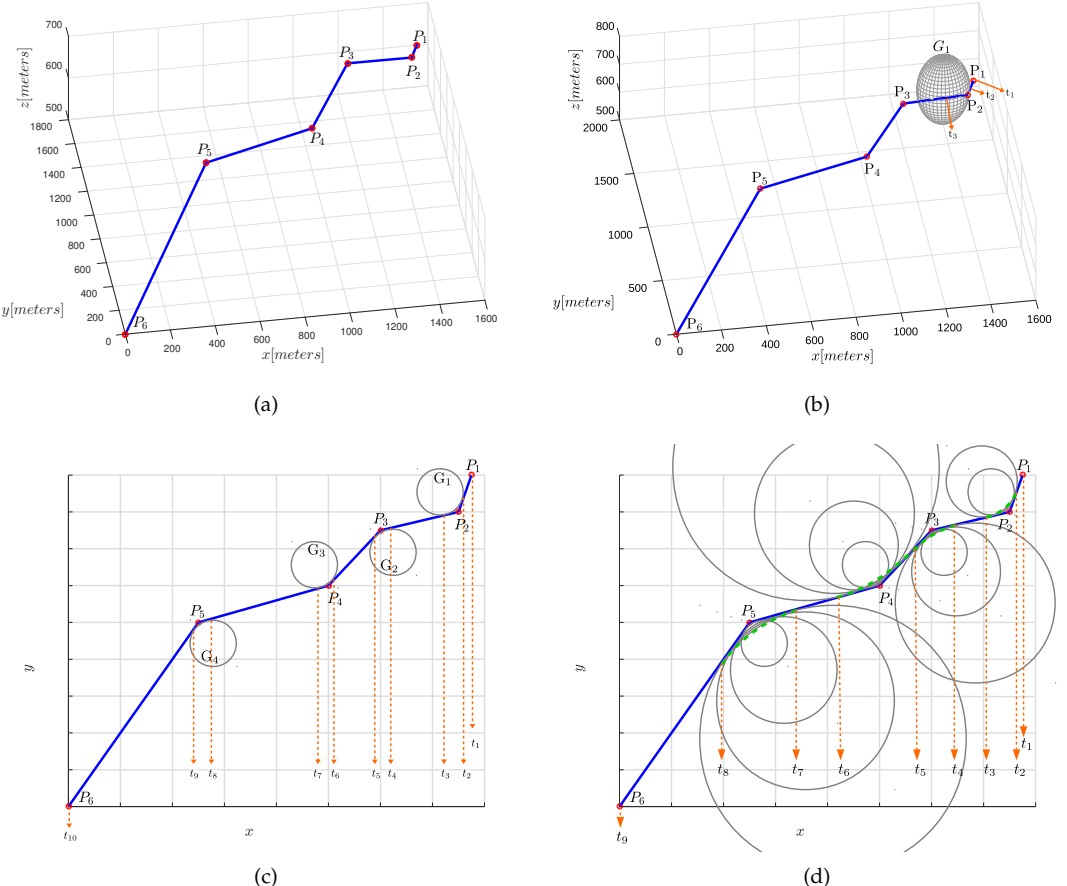

**Figure 2.** Smooth path planning problem. The red dots represent $\rho$ and the blue line is the path made up of straight-lines $\boldsymbol{L}(t)$. (**a**) Result of path planning with $\rho$ collision-free points. (**b**) Definition of a sphere with a relation of the $Rp$ minimum. (**c**) Set of $G_i$ over $\rho$. (**d**) Example of optimal smooth trajectory, with optimized $\kappa$ and $\tau$ represented by dotted green lines in orthogonal view.

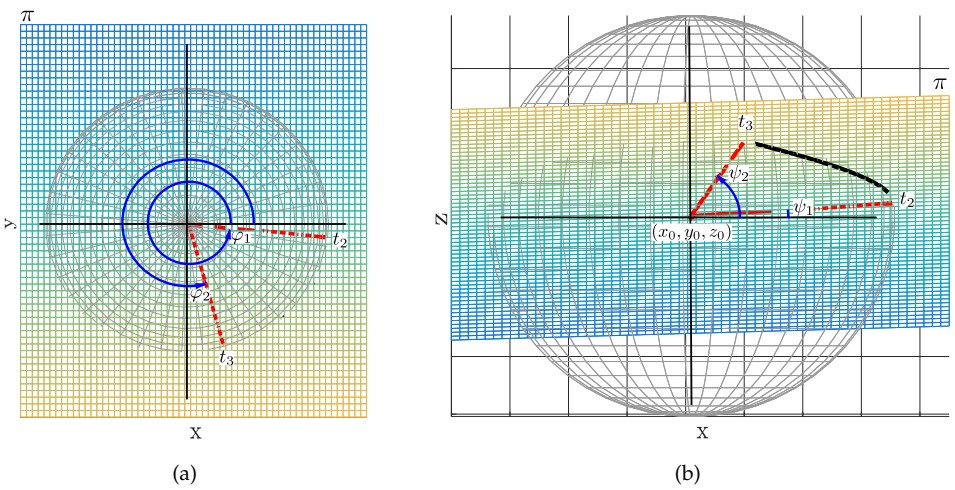

**Figure 3.** Spherical smooth path, where the black line shows the curve in a segment $S_i$ along the plane $\pi_i$. The red lines show the union from the center of coordinates of the sphere to the points of intersection between the sphere and the $\pi_i$ plane resulting in the spherical semicurve. (**a**) View perpendicular to the horizontal plane $(x, y)$. (**b**) Perpendicular view to the vertical plane $(x, z)$.

**Remark 1.** *If the plane $\pi_i$ is parallel to the horizontal plane $(x, y)$ of the environment, then $\tau = 0$ because the movements of the UAV will be horizontal. In the same way, if $\pi_i$ is parallel to the vertical plane $(x, z)$ of the environment, then $\kappa = 0$.*

However, before applying Equation (17), it is necessary to determine the location of the points $(x_0, y_0, z_0)$ so that $G_i$ is tangent at a point on its surface with $L(t)$, as shown in Figure 2b on the points ($t_2$ and $t_3$). Nevertheless, it should be noted that there is an angle between each pair of $L(t)$, and this leads to $G_i$ approaching or moving away from the lines and their tangent points. Therefore, the geometric analysis applied to arrive at an optimal solution is detailed below.

First, a vector direction in space can be defined as $\overrightarrow{v} = p - q : p \wedge q \in \mathbb{R}^3$. Therefore, starting from the known data $\rho = [P_1, \cdots, P_n]$, taking Figure 2 as an example, where it is assumed that the collision-free initial points are ($P_i : \{i = 1, \cdots, 3\}$), a first set of two vectors is defined as:

$$\left.\begin{array}{l} \overrightarrow{u}_i = p - q : p = P_{(i+1)}, q = P_{(i)} \\ \overrightarrow{v}_i = p - q : p = P_{(i+1)}, q = P_{(i+2)} \end{array}\right\}, i = 1. \tag{18}$$

Just like a perpendicular vector from $\overrightarrow{u}_i$ to $\overrightarrow{v}_i$, denoted as $\overrightarrow{\eta}$, the normal vector is defined as:

$$\overrightarrow{\eta} = \overrightarrow{u}_i \times \overrightarrow{v}_i. \tag{19}$$

Consequently, the parametric equation of the $\pi_i$ plane containing three points is defined as:

$$\pi_i = \left.\begin{bmatrix} (x - p_x) \\ (y - p_y) \\ (z - p_z) \end{bmatrix} * \begin{bmatrix} \overrightarrow{\eta} \end{bmatrix}\right\} : p_x = P_{(i+1)_x}, p_y = P_{(i+1)_y}, p_z = P_{(i+1)_z}. \tag{20}$$

In the same way, the Euclidean distance defined between two points $p$ and $q$ is given by

$$d(p, q) = \sqrt{\sum(p - q)^2}. \tag{21}$$

Therefore, two distances can be defined as $du_j : p = P_{(i+1)}, q = P_{(i)}$ and $dv_j : p = P_{(i+1)}, q = P_{(i+2)}$. Finally, the angle between $\overrightarrow{v}_i$ and $\overrightarrow{u}_i$ is defined by:

$$\angle(\overrightarrow{u}_i, \overrightarrow{v}_i) = \phi_i = \tan^{-1} \frac{||\overrightarrow{u}_i \times \overrightarrow{v}_i||}{\overrightarrow{u}_i \cdot \overrightarrow{v}_i}. \tag{22}$$

Therefore, with Equation (22) and $Rp$ known, the tangential points at the lines $L(t)$ can be located at a distance defined as:

$$\sigma_i = \frac{Rp}{\phi_i / 2}. \tag{23}$$

Thereby, two spatial points defined as $pUi_i$ and $pUg_i$ located in the direction of the vector $\overrightarrow{u}_i$ provide that $Pi = P_{i+1}, Pg = P_i$ and $d(p, q) = du_i$; thus

$$\begin{array}{l} \gamma = \sigma_i / d(p, q) \\ pUi_j = (Pi - Pg) * \gamma + Pi \\ pUg_j = -(Pi - Pg) * \gamma + Pi. \end{array} \tag{24}$$

In the same way, two points $pVi_i$ and $pVg_i$ can be defined in the vector direction $\overrightarrow{v}_i$, so long as $Pi = P_{i+1}, Pg = P_{i+2}$ and $d(p, q) = dv_i$, according to Equation (24). Hence, the perpendicular bisector of $pUi_i$ and $pVi_i$ on the plane $\pi_i$ determines the center of the sphere $(x_0, y_0, z_0)$. Figure 4a shows the application of the Equations (18)–(24), which can be repeated throughout the successive collision-free points $\rho$ (this first Algorithm 1 is summarized in pseudocode). Between the centers of the spheres

$(x_0, y_0, z_0)$ and the points of intersection with $L(t)$ are the displacement angles $\varphi$ and $\psi$ of the segment $S_i$, as can be seen in the Figure 3, where $pUi_i = t_2$ and $pVi_i = t_3$.

**Remark 2.** *Regardless of the angle condition produced by the pair of straight lines $L(t)$ denoted in the Equation (22), the angle formed between the points of intersection on the sphere $G_i$, seen from its center towards the vertical or horizontal component, does not exceed* $90°$ *in any case; that is,* $(0° < \varphi < 90°)$ *and* $(0° < \psi < 90°)$.

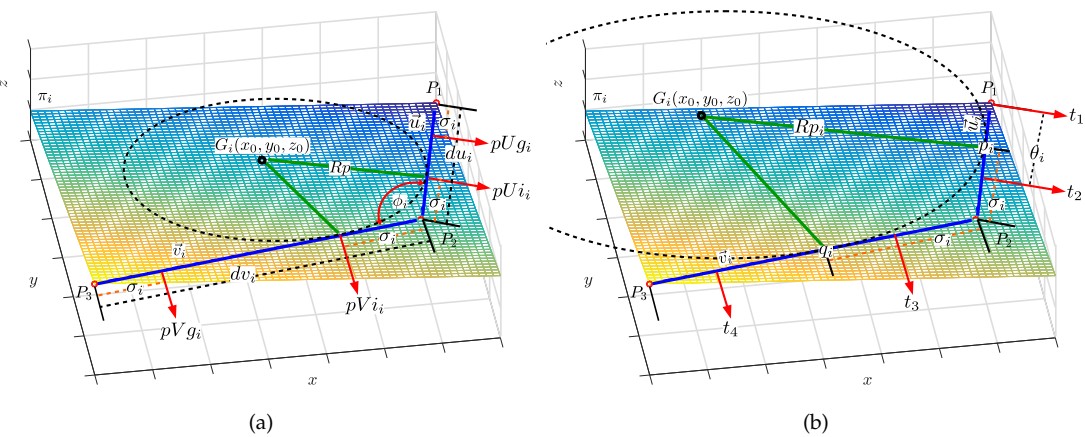

(a)　　　　　　　　　　　　　　　　　　　　　　(b)

**Figure 4.** Description of the methodological basis. The circle of black dots shows the osculating sphere $(oS)$; the green line is the radius of turn $Rp_i$. (**a**) Sphere location $G_i$ with minimum turning radius. (**b**) Sphere location $G_i$ with upper turning radius displaced within the bounds given by $[t_1, t_2]$ and defined by the value of $\theta_i$.

---

**Algorithm 1** First set of $G_i$

---

1: $\rho = [P_1, \cdots, P_n] \rightarrow$ Path planning problem result.
2: $Rp =$ minimum sphere radius.
3: $k = 1$ counter for the set of intervals $t$.
4: **for** $i = 1 : n - 2$ **do**

5: 　　$(\overrightarrow{u}_i, \overrightarrow{v}_i) =$ vectors .directions;
6: 　　$\phi_i =$ angle.betweeLines$(\overrightarrow{u}_i, \overrightarrow{v}_i)$
7: 　　$\sigma_i =$ distance.intersectionPoint$(Rp, \phi_i)$;
8: 　　$(pUi_i, pUg_i) =$ intersectPoint.$(P_{i+1}, P_i, \sigma_i)$;
9: 　　$(pVi_i, pVg_i) =$ intersectPoint.$(P_{i+1}, P_{i+2}, \sigma_i)$;
10: 　$G_i(x_0, y_0, z_0) =$ bisectPoint$(pUi_i, pVi_i, \pi_i)$;
11: 　**if** i==1 **then**

12: 　　　$t_k = P_1$;
13: 　　　$t_{k+1} = pUi_i$;
14: 　**else**

15: 　　　$t_k = pUg_j$;
16: 　　　$t_{k+1} = pUi_i$;
17: 　**end if**
18: 　**if** i==n-2 **then**

19: 　　　$t_{k+2} = pVi_i$;
20: 　　　$t_{k+3} = P_n$;
21: 　**end if**
22: 　$k = k + 2$;
23: **end for**

---

Algorithm 1 summarizes the geometric procedure followed. In line 4, a loop is started that performs through all the collision-free points marked by *rho*. From line 5 to line 9, the necessary

computations are made to determine the center of $G_i$ (line 10). The definitions of the intervals containing the $S$ and $L$ segments are in lines 12, 13, 15, 16, 19 and 20.

The described process shows the geometric analysis for the location of the set of spheres $G_i$ defined with constant radius $Rp$, as can be seen in Figure 2c. In addition, there is a set of four segments $S$ and another set of five segments $L$, being segments $S$—those comprised by the intervals $[t_2, t_3]$, $[t_4, t_5]$, $[t_6, t_7]$ and $[t_8, t_9]$, and the segment $L$ included by the intervals $[t_1, t_2]$, $[t_3, t_4]$, $[t_5, t_6]$, $[t_7, t_8]$ and $[t_9, t_{10}]$. The goal now is to increase the radius $Rp$ in each segment, so that the values of $\kappa$ and $\tau$ along the curve are minimized, and the solution is to increase the radius $Rp_i$ in each $G_i$.

The solution adopted in this work is to move the intersection point of each sphere $G_i$ in the direction of the adjacent segment $L(t)$. Consequently, $G_i : i = 1$ approximates symmetrically to the intervals $t_1$ and $t_4$, $G_i : i = 2$ makes the corresponding approximation to the intervals $t_3$ and $t_6$, etc. Therefore, in Figure 4b, the segments $L(t)$ can be seen adjacently to $G_i : i = 1$, denoted as $[t_1 \equiv P_1, t_2 \equiv pUi_j, t_3 \equiv pVi_j, t_4 \equiv pVg_j]$. Thus, between the adjacent intervals $[t_1, t_2]$ a vector is defined $\overrightarrow{u}_i = t_2 - t_1$, and associated with this vector is a spatial point $p_i$ defined by the parametric equation:

$$\left. \begin{array}{l} p_{i_x} = t_{2_x} + \theta_i * \overrightarrow{u}_{i_x} \\ p_{i_y} = t_{2_y} + \theta_i * \overrightarrow{u}_{i_y} \\ p_{i_z} = t_{2_z} + \theta_i * \overrightarrow{u}_{i_z} \end{array} \right\} , 0 \le \theta_i \le 1, \tag{25}$$

where $\theta_i$ defines the space point $p_i$ along $\overrightarrow{u_i}$ and within the intervals $[t_1, t_2]$. Therefore, the value of the distance $\sigma_i$ from $P_{i+1}$ to $p_i$ is defined according to the equation on (21), being $p = P_{i+1}$ and $q = p_{i+2}$. A symbolic space point is defined $q_i$ between the intervals $[t_3, t_4]$ with direction $\overrightarrow{v_i} = t_3 - t_4$ at the same distance $\sigma_i$. Then $\sigma_i$ also has the angle $\phi_i$, and according to the Equation (22) it is possible to define a new $Rp_i$ according to Equation (23), which would have a higher radius value. Finally, the perpendicular bisector of $p_i$ and $q_i$ on the plane $\pi_i$ determines the center of $G_i(x_0, y_0, z_0)$ (see Figure 4b). Therefore, as the center of $G_i$ is defined, the Equation (17) defines the segment $S_i$—and over this segment we find lower values of $\kappa$ and $\tau$ according to the Equations (12) and (13).

Multiobjective Problem Definition (MOP)

Given Equation (25), it is important to note that any value of $\theta_i$ between 0 and 1, defines a space point between the interval $[t_1, t_2]$. In the same way, it is important to mention that within the boundaries of $\theta_i$, there is an infinite number of space points with an infinite number of radius $Rp_i$ and its corresponding infinite number of $G_i$, with which its corresponding segments $S_i$, can be built.

Therefore, to obtain an optimal solution, a multiobjective problem (MOP) is solved using evolutionary algorithms [70] based on the concept of $\epsilon$-dominance [71]. To do this, it is necessary to define the decision variables, the initial conditions of the process, the constraints of the MOP and the index vector to be optimized to represent the Pareto front. If it is assumed that the number of spheres $G_i$ is equal to $m$, and the number of objectives for each $G_i$ is equal to two, then $J^{ideal}(\theta) = [J_1(\theta), J_2(\theta), \cdots, J_{2*m}(\theta)]$ is the objectives vector, where $J_i$ denotes the $i^{th}$ objective. Consequently, $J_i^A = \min(\kappa(\theta_i))$, $J_i^B = \min(\tau(\theta_i)) \in G_i : [i = 1, \cdots, m]$, where $J_i^A$ and $J_i^B$ depend on the decision variables vector $\theta$. Assume $D$ as a decision space within a subset $\mathbb{R}^D$, where $\theta$ is the decision variable vector composed of a set of $\theta_i$ for all $i \in 1 < i < m$, where $\theta_i$ is $[0, 1]^D$. Consequently, the MOP problem can be stated as:

$$\min_{\theta \in D} [J_i^A(\theta), J_i^B(\theta)]_{1 \times (2*m)}, \ \forall i \in 1 \le i \le m. \tag{26}$$

where

$$J_i^A = \frac{\| S_i'(t) \times S_i''(t) \|}{\| S_i'(t) \|^3},$$

from Equation (12)

$$J_i^B = \frac{S_i'(t) \cdot [S_i''(t) \times S_i'''(t)]}{\|S_i'(t) \times S_i''(t)\|^2},$$

from Equation (13)

$$\theta = [\theta_i]_{1 \times m}, \ \forall i \in 1 \leq i \leq m$$

subject to:

$$S_i(t) = \begin{cases} S_{i_x} = & x_0 + Rp_i * \sin(\psi) * cos(\varphi) \\ S_{i_y} = & y_0 + Rp_i * \sin(\psi) * sin(\varphi) \\ S_{i_z} = & z_0 + Rp_i * \cos(\psi) \end{cases}$$

from ec. (17)

$$Rp_i = \sigma_i * (\phi_i/2)$$

from Equation (23)

$$\sigma_i = \sqrt{\sum (p_i - P_{i+1})^2}$$

from Equation (21)

$$p_i = \begin{cases} p_{i_x} = t_{2_x} + \theta_i * \overrightarrow{u}_{i_x} \\ p_{i_y} = t_{2_y} + \theta_i * \overrightarrow{u}_{i_y} \\ p_{i_z} = t_{2_z} + \theta_i * \overrightarrow{u}_{i_z} \end{cases}$$

from Equation (25)

$$\theta_i \in [0,1]^D.$$

In summary, the aim is to find an optimal 3D smooth curve that minimizes $\kappa$ and $\tau$ in each of the possible $S_i$. It is important to mention that the adjacent spheres $G_i$ can grow into each other, until a maximum of $q_i \in \overrightarrow{v_i} \equiv p_{i+1} \in \overrightarrow{u_i}$, which implies a decrease in the total set of segments, as described Figure 2d, where the green dotted line shows the set of $S_i$ segments belonging to $G_i$.

An example of reconstruction according to the response $\Theta_P^*$ can be seen in Figure 2d, where the $S$ reconstruction is made in four segments, defined by the boundaries $[t_2, t_3]$, $[t_4, t_5]$, $[t_5, t_6]$ and $[t_7, t_8]$; the $S$ segments belonging to $C(t)$ are defined according to Equation (17).

In contrast, and with reference to Figure 2d, the $L$ segments are defined by the rest of the boundaries, those boundaries being $[t_1, t_2]$, $[t_3, t_4]$, $[t_6, t_7]$ and $[t_8, t_9]$.

### 4.2. Definition of Straight-Line Segment

A segment path of $L$ in a straight-line can be described by two points in the Euclidean space. Figure 2d shows an example of an $L$ segment defined by the $[t_1, t_2]$ points, where the direction of the line is given by the flight path of the UAV. Therefore, $\overrightarrow{u}$ (Figure 5) is a unit vector that points in the direction of the desired orientation, and with $d$ defined as the distance between $t_1$ and $t_2$ according to Equation (21). Therefore, the $L$ segments will be described, in general, as:

$$L(t) = \{r \in \mathbb{R}^3 : r = (t_1 - t_2) * \gamma - t_1\} \rightarrow 0 \leq \gamma \leq d. \tag{27}$$

Finally, the interpolation of $S$ and $L$ build a final 3D smooth curve on the plane $(x, y, z)$.

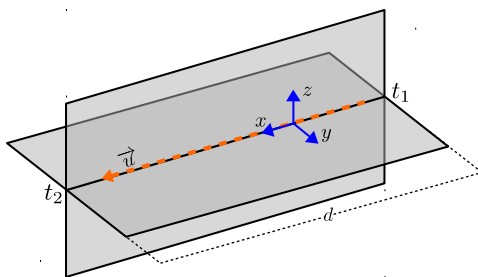

**Figure 5.** Straight-line segment.

## 5. Experiments and Results

This section presents the results of the computer simulation using Matlab/Simulink software and flightGear flight simulator for graphical visualization.

In this section we analyze five scenarios in 3D space, similar to the methodology proposed in [44]. Recursive rewarding modified adaptive cell decomposition (RR-MACD) splits the 3D environment like a discrete mesh of collision-free voxels. In particular, it places the UAV within an initial voxel and determines which of the adjacent voxels is the most suitable to make a displacement. To determine the best choice of displacement, the set of adjacent voxels have a set of associated constraints (conditions such as distance, vertical, and horizontal movement angles, battery consumption, etc.) to be satisfied before performing a discrete displacement. The voxel that minimizes the total effort and satisfies the constraints will be the next collision-free point. The RR-MACD methodology gives two sets of results based on the defined constraints. The results presented in [44] are shown in summary form in Table 1, where the first column shows the scenario number. The second column shows RR-MACD with four constraints and the RR-MACD with 10 constraints in the third column shows the conditions to solve the path planning problem. The 3D control points reflected in Table 1, $\rho_x(F) \approx \rho$, are the starting points for analyzing the method described in this paper for generating 3D smooth curves. Finally, it is important to note that the algorithms have been run on an Intel(R) Core(TM) i7-4790 3.60 GHz CPU (Manufacturer: Gigabyte Technology Co., Ltd., Model: B85M-D3H) with 8Gb RAM and S.O. Ubuntu Linux 16.04 LTS. The algorithms were programmed in MATLAB version 9.4.0.813654 (R2018a).

**Table 1.** 3D path planning results. The number of free-collision voxels within a defined environment is indicated as $S_{free}$, while the number of discrete 3D nodes built by [44] is denoted by $\rho_x(F)$ (3D final path).

| Env. | RR-MACD 4 Constraints | | RR-MACD 10 Constraints | |
|---|---|---|---|---|
| | # $S_{free}$ | # $\rho_x(F)$ | # $S_{free}$ | # $\rho_x(F)$ |
| # 1 | 115 | 18 | 202 | 27 |
| # 2 | 27 | 8 | 35 | 10 |
| # 3 | 19 | 6 | 16 | 7 |
| # 4 | 11 | 6 | 51 | 10 |
| # 5 | 19 | 7 | 35 | 10 |

It is important to mention that the characteristics of the UAV assumed in the experiments have been taken from [65], whose study has been carried out on a fixed wing UAV model *kadett 2400*, represented by six states $(x, y, z, \phi, \theta, \psi)$, where the first three states define the position vector of the UAV's global coordinate system, located at the origin of its center of gravity. The last three are the Euler angles of roll, pitch and yaw respectively, which define the orientation of the UAV.

Finally, the aim of the simulations is for the UAV to maintain its continuous flight at a defined constant speed of 18 m/s, within an established minimum radius of curvature $Rp = 33$ m to achieve smooth behavior and without maneuvers that could endanger the integrity of the aircraft.

It is important to remember that because the number of $\rho_x(F) = [P_1, \cdots, P_n]$ collision-free points is greater than five, a proper visualization method is essential to the decision making process for the final solution. Thus, the graphical representation method called level diagram [72] has been used, which consists of representing each objective and each design parameter in separate diagrams, all of which are synchronized with their $y$ axis. Synchronization is made with the normalized distance of each point from the Pareto front to the ideal point. Therefore, with a brief training this representation offers a good visual understanding of the compromise reached on the Pareto front.

### 5.1. Bézier

To compare the results, an approximation of curves has been applied using Bézier [68], which enables the generation of trajectories for non-holonomic systems through a set of discrete control points, where the curve in a multidimensional environment can be described as:

$$B(t) = \sum_{i=0}^{n} P_i b_{i,n}(t), \ t \in [0,1]$$

$$b_{i,n}(t) = \binom{n}{i} t^i (1-t)^{n-1}, \ i = 0, \cdots, n$$

(28)

Bernstein-Bézier generates a finite vector of points belonging to the curve and guarantees to go through the first and last control points (translated as $\rho = [P_1, \cdots, P_n]$) and remaining inside the convex envelope.

### 5.2. Application Example

To represent visual and numerical results from Table 1, the results for the #3 environment with RR-MACD with four constraints are detailed below. As in this example, the total number of $P_i$ equals 6, and this means that the decision criteria of Equation (26) $\rightarrow m = 4$. Therefore, there are four values of $\kappa$ and four values of $\tau$; i.e., $\Theta_p^* = (J_1(\theta_1) = \kappa_1, J_3(\theta_2) = \kappa_2, J_5(\theta_3) = \kappa_3, J_7(\theta_4) = \kappa_4, J_2(\theta_1) = \tau_1, J_4(\theta_2) = \tau_2, J_6(\theta_3) = \tau_3$ and $J_8(\theta_4) = \tau_4$), as can be seen in Figure 6.

It is important to mention that the interpolation of segments $S$ and $L$ builds a set of 3D smooth curves, all of which are possible solutions. It is, therefore, necessary to address a decision stage (decision maker (DM)) that selects one of them; i.e., a point on the Pareto front. In this work, the selection criteria based on the shortest distance to the ideal point have been used. Figures 6 and 7 show the selected point in red from $J(\Theta_p^*)$ and $\Theta_p^*$, which have been selected using the $\infty$-norm standard.

Figure 8a shows the build of the 3D smooth curve $C(t)$, while Figure 8b shows the best optimization in terms of curvature $\kappa$ and total torsion $\tau$, according to Equations (12) and (13). It is important to note that the mathematical mean of the values of the geometric variables $\kappa$ and $\tau$ tends to be low. However, in some particular cases an increase is detected due to the change in direction of the flight; i.e., when the UAV is terminating an $S$ segment in one direction and another $S$ segment begins in the opposite direction. The curve generated by Bézier $B(t)$ is shown as a yellow line, a more direct route can be seen between the init and goal point. However, this curve is very close to the bottom obstacle. To solve this, different authors propose modifying the control points $\rho$, or adding new points within the points established initially, as remarked in Section 1.

Figure 9 shows the set of four additional examples from Table 1, in order to represent the functionality of the algorithm. Similarly, it is important to note that the number of $\rho$ was different in each experiment, as were the altitudes, which guaranteed movement and 3D planning. It is also important to mention that the first environment shown in Figure 9a has smaller flight dimensional

characteristics, so the turning radius in this example was set at $Rp = 3$ m with an average flight speed of 1.7 m/s.

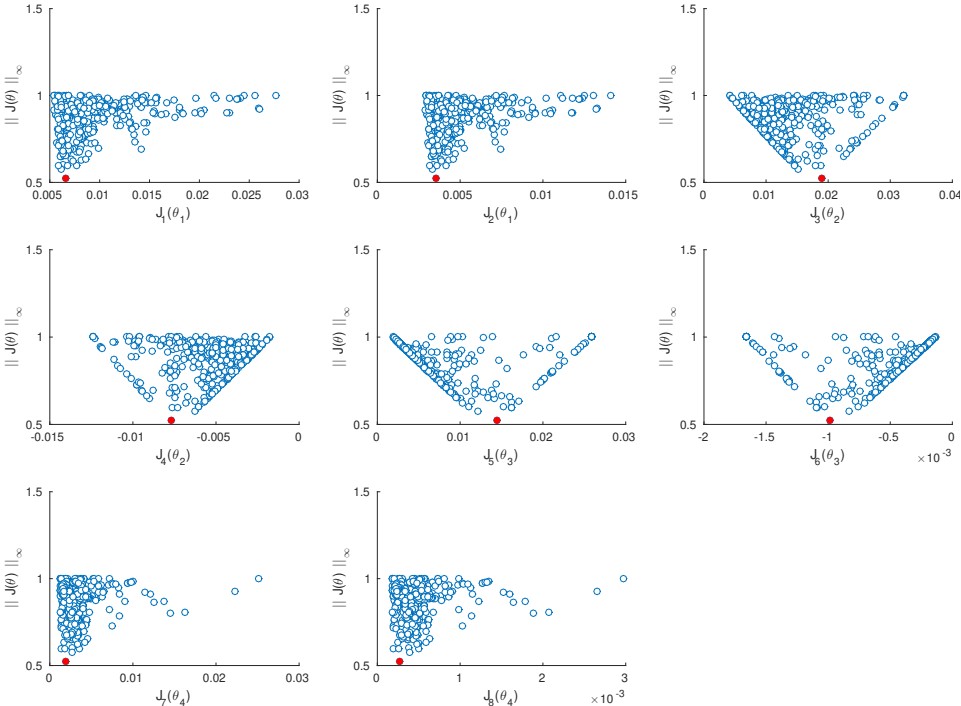

**Figure 6.** Representation of Pareto front using $\infty - norm$. The $J$ pair sub-indices represent the $\kappa$ values in each $S_i$ segment, while the $J$ odd sub-indices represent the $\tau$ values in the same $S_i$ segments. Targets closest to $J^{ideal}$ are shaded in red circles.

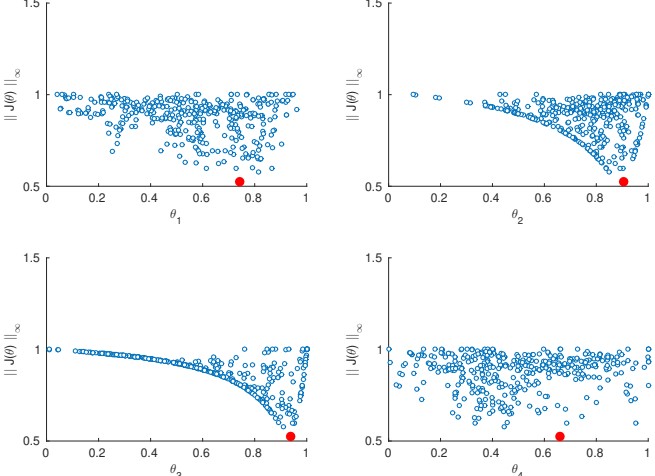

**Figure 7.** Representation of Pareto's optimal parameters. Targets closest to $J^{ideal}$ are shaded in red circles.

To describe the different groups of trajectories generated by $L(t)$, $C(t)$ or $B(t)$ displayed in Figure 9, Table 2 shows the flight results from the init to goal point in terms of distances; according to one of the results of each environment set by Table 1. It can be seen that the set of greater distances corresponding to the path in the form of the straight line marked by $L(t)$, $C(t)$ reduces the distance by $L(t)$. As $B(t)$ makes an approximation (as a mathematical expression) between the set of $\rho$ of each environment, its route is the shortest. The column *"EAA Error (meters)"* shows the approximate absolute error $EAA = \frac{1}{n} \sum_{i=1}^{n} |A - B|$, where $A = L(t)$ and $B = C(t) \wedge B = B(t)$. Therefore, the results

of the column *EAA Error (meters)* show a greater approximation by $C(t)$ and which results in a better avoidance of obstacles.

**Table 2.** Flight distance of the curves. The column *"Flight distance (meters)"* shows the distance in meters at the initial and final free collision points marked by $\rho$. The column *"EAA Error (meters)"* shows the average error in meters along the trajectories.

| Env. | Flight Distance [Meters] | | | EAA Error [Meters] | |
|------|------|------|------|------|------|
| | *L(t)* | *C(t)* | *B(t)* | *L(t) vs C(t)* | *L(t) vs B(t)* |
| *#1* | 182.929355 | 174.002834 | 148.911388 | 0.622684 | 3.248545 |
| *#2* | 1728.757868 | 1610.781941 | 1453.060601 | 17.234613 | 41.453691 |
| *#3* | 1863.391222 | 1721.505017 | 1526.055284 | 14.600159 | 56.678212 |
| *#4* | 1936.078758 | 1860.263202 | 1772.944453 | 9.871725 | 36.617234 |
| *#5* | 1873.814514 | 1839.965587 | 1743.723244 | 9.891240 | 36.614752 |

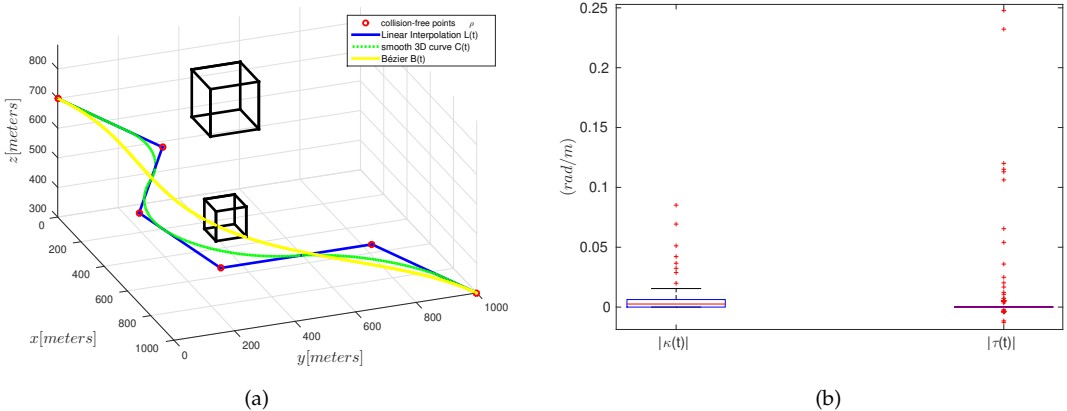

(a)                                                            (b)

**Figure 8.** Example of a 3D environment cluttered of obstacles. (**a**) Reconstruction of 3D trajectories, where the obstacles are the black boxes; the red circles show the collision-free points; and the green line shows the final 3D trajectory $C(t)$. The yellow line represents the Bézier $B(t)$ curve. (**b**) Geometric averages of the variables $\kappa$ and $\tau$ of the final path.

Similarly, Table 3 shows a set of results referring to the five environments analyzed. The averages of $\kappa$ and $\tau$ generated along each smooth curve shows that $B(t)$ exceeds $C(t)$. However, in the first environment there is a collision caused by the $B(t)$ curve.

**Table 3.** Average results of $\kappa$ and $\tau$ along the curves $C(t)$ and $B(t)$. The column "Collision" indicates collision (x) or no collision (o) of a curve against an obstacle.

| Env. | Curve | $\kappa$ | $\tau$ | Collision |
|------|-------|----------|--------|-----------|
| *#1* | *C(t)* | 0.157961 | 0.185973 | o |
| | *B(t)* | 0.019513 | 0.092539 | x |
| *#2* | *C(t)* | 0.007138 | 0.159732 | o |
| | *B(t)* | 0.001082 | 0.006652 | o |
| *#3* | *C(t)* | 0.004556 | 0.185806 | o |
| | *B(t)* | 0.001068 | 0.004442 | o |
| *#4* | *C(t)* | 0.003445 | 0.574121 | o |
| | *B(t)* | 0.000812 | 0.003332 | o |
| *#5* | *C(t)* | 0.004515 | 0.135183 | o |
| | *B(t)* | 0.000643 | 0.004253 | o |

Finally, the results produced by the simulation of the UAV kaddet 2400 matlab/simulink/ flightGear over the environment #3 are shown in Figure 10. The geodesic coordinates of Figure 10a are

expressed in decimal degrees; in this example the flight starts with an altitude of 500.4 m, and after the maneuvers performed by the UAV, it reaches a new altitude of 603.1 m. Figure 10b shows the UAV model in flight using FlightGear Simulator as visualization platform.

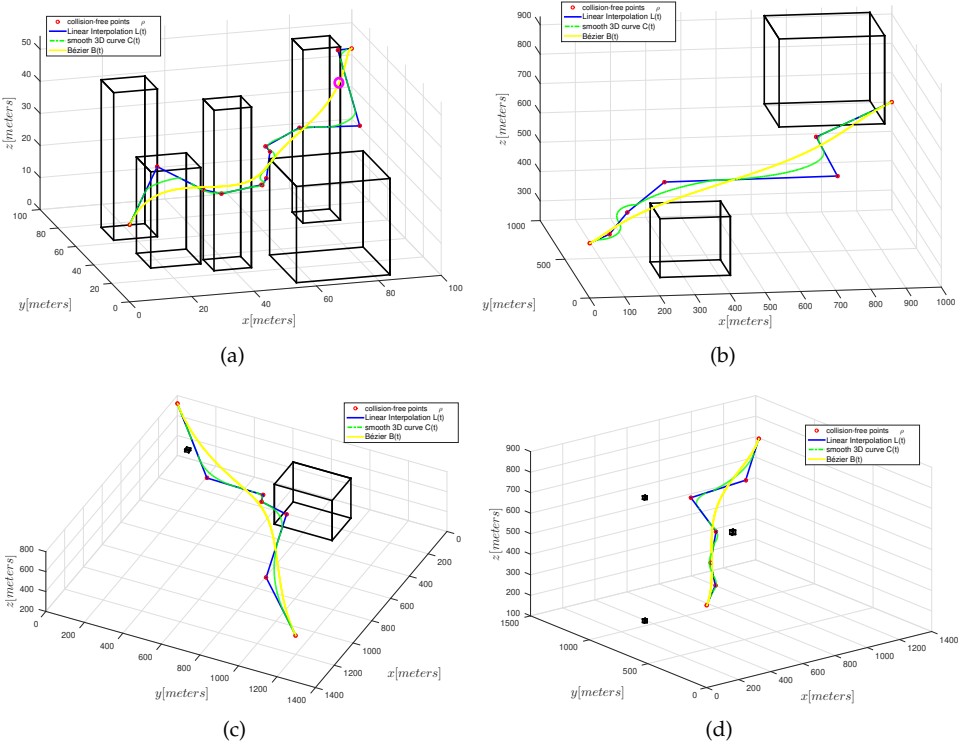

(a)

(b)

(c)

(d)

**Figure 9.** Additional 3D environment experiments in which the obstacles are the black boxes and the green line shows the final 3D trajectory $C(t)$; the yellow line represents the Bézier $B(t)$ curve. (**a**) (Table 1—Environment #1.) It represents an unstructured environment with different buildings, where a collision between $B(t)$ and a building (collision marked as a circumference of magenta color) can be seen. (**b**) (Table 1—Environment #2.) 3D environment with two obstacles of different sizes. (**c**) (Table 1—Environment #4.) 3D environment with two obstacles of different sizes. (**d**) (Table 1—Environment #5.) 3D environment with three small aerial obstacles.

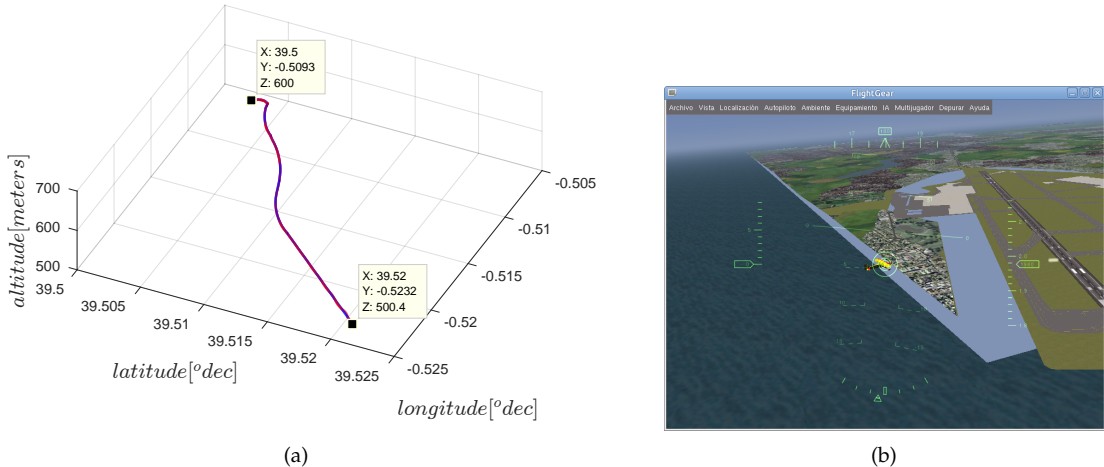

(a)

(b)

**Figure 10.** Simulation of the UAV flight simulink, where the starting point is $[39.52, -0.5232, 500]$ and the target point is $[39.50, -0.5093, 600]$. (**a**) Flight of the UAV, where the blue line is the path calculated from the process described and the red line is the actual path of the UAV. (**b**) View from a model perspective.

## 6. Conclusions and Future Works

This paper describes an approach to the generation of a continuous 3D smooth path that enables consideration of the operational constraints of fixed-wing UAVs.

Firstly, the document describes the formulation of the problem by defining two types of segments within the trajectory: the *S* segments as a set of sphere segments that ensure a continuous and minimum curvature profile, and then the definition of *L* segments that generally connect *S*.

Next, we proposed the resolution of an MOP problem to obtain the numerical values of the parameters of the trajectory, given that the problem has infinite feasible solutions. When solving an MOP problem, the DM stage is essential to finally select the desired point from the Pareto set of optimal solutions.

It is important to remember that with methods such as classic Bézier or B-splines curves, you can define the number of samples along the path. However, the distance measured between one point and the next is not the same or even close (the difference can be large). These types of curves are useful in relatively simple environments (few obstacles); however, as the number of obstacles grows, the control points increase due to trajectory planning. Consequently, the construction of the curve can cause collisions. This work enables a constant approach distance between pairs of contiguous points.

The kinematic constraints of UAVs have been considered in this work, in the same way that dynamic constraints could be calculated mathematically. However, an important consideration that can enhance the generation of new trajectories in a new job is to increase the optimization criterion by improving variables such as energy consumption or incomplete data in dynamic environments.

Connected with the results shown in this paper, several future works arise. For example, integration of dynamic obstacles (UAVs swarms or other aircraft systems) into the flying area. From the optimization point of view, the proposal can be improved by taking into account dynamic constraints (i.e, inertia, wind disturbances, torque forces, etc.) into the MOP problem. Similarly, new cost indexes as flight time or/and spent energy could be added to the optimization problem. Finally, implementation of the proposed algorithm under real conditions (UAV in an outdoor environment) and its application to different uses (such as satellite trajectory generation [73]) will be investigated.

**Author Contributions:** Conceptualization, F.S.; formal analysis, F.S., J.S., S.G.-N. and R.S.; methodology, F.S.; supervision, J.S., S.G.-N. and R.S. All authors have read and agreed to the published version of the manuscript.

**Funding:** The authors would like to acknowledge the Spanish Ministerio de Ciencia, Innovación y Universidades for providing funding through the project RTI2018-096904-B-I00 and the local administration Generalitat Valenciana through projects GV/2017/029 and AICO/2019/055. Franklin Samaniego thanks IFTH (Instituto de Fomento al Talento Humano) Ecuador (2015-AR2Q9209), for its sponsorship of this work.

**Acknowledgments:** The authors would like to thank the editors and the reviewers for their valuable time and constructive comments.

**Conflicts of Interest:** The authors declare no conflict of interest.

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
