# Peer review of "Smooth 3D Path Planning by Means of Multiobjective Optimization for Fixed-Wing UAVs"

_electronics, doi:10.3390/electronics9010051_

Round 1
Reviewer 1 Report
The paper reports on actual and interesting topic of development effective optimization algorithms for generating flight paths for UAVs.
The major part of the paper describes the multiobjective optimization algorithm taking into account non-holonomic constraints due to manoeuvrability ov the UAV. But unfortunately, there is no information on constraints caused by increasing amount of UAVs. The advanced algorithms of 3D path planning should take into account that there could be dozens or even hundreeds (in future) another UAVs flying. Please, add some possible functions of UAVs the paper focused on.
In order to improve the paper the authors should make some edition (fix typos):
1) Line 31: VA should be changed by AV.
2) Line 33: "This paper focuses on the generation of smooth ..." (add "of").
3) Line 106: "...X that generates an optimum..." (optimum instead of optimal).
4) Line 114: "... the lower and upper boundaries..." (the order of "lower" and "upper" should be the same as of corresponding symbols).
5) Line 183: "possible values of k and t..." (k instead of "kappa").
Author Response
We would like to thank the reviewers for their time and efforts on reviewing our submission. It is a pleasure to receive support from experienced professionals who express such accurate opinions. We have modified the manuscript with the aim of solving each of reviewer’s comments and suggestions, emphasizing the changes on the modified manuscript.
Response to Reviewer 1
Point 1: The major part of the paper describes the multiobjective optimization algorithm taking into account non-holonomic constraints due to manoeuvrability ov the UAV. But unfortunately, there is no information on constraints caused by increasing amount of UAVs. The advanced algorithms of 3D path planning should take into account that there could be dozens or even hundreeds (in future) another UAVs flying. Please, add some possible functions of UAVs the paper focused on.
Response 1: The comment has been taken into account and added in the section future works - Lines (437–443):
“Connected with the results shown in this paper, several future works arise. For example, integration of dynamic obstacles (UAVs swarms or other aircraft systems) into the flying area. From the optimization point of view, the proposal can be improved taken into account dynamic constraints (i.e, inertia, wind disturbances, torque forces, etc.) into the MOP problem. Similarly, new cost indexes as flight time or/and spent energy could be added to the optimization problem. Finally, implementation of the proposed algorithm under real conditions (UAV in an outdoor environment) and application to different uses (such as satellites trajectories generation [82]) will be investigated.”
Point 2: Please, add some possible functions of UAVs the paper focused on.
Response 2: (Lines 34 -36 has been added)
“For these reasons, fixed-wing UAVs are suitable for use in terrain mapping applications for later action by the security forces, as well as in search and rescue tasks, both for the detection of people and provision of first aid.”
Point 3: In order to improve the paper the authors should make some edition (fix typos):
Response 3: The text you mentioned has been modified as specified in the following lines.
1) Line 31: VA should be changed by AV. (modified in line 31)
2) Line 33: "This paper focuses on the generation of smooth ..." (add "of"). (modified in line 33)
3) Line 106: "...X that generates an optimum..." (optimum instead of optimal). (modified in line 112)
4) Line 114: "... the lower and upper boundaries..." (the order of "lower" and "upper" should be the same as of corresponding symbols). (modified in line 120)
5) Line 183: "possible values of k and t..." (k instead of "kappa"). (modified in line 189)

Reviewer 2 Report
Smooth 3D path planning by means of Multi-Objective optimization for fixed-wing UAVs
Electronics,
In this paper, the authors developed an approach to the generation of a continuous 3D smooth path that enables consideration of the operational constraints of UAVs. The manuscript is well written and organized and has reasonable technical merit. It could be considered for accept in Electronics after the following issues and questions are addressed and explained.
The key parts of the approach in this work is the set of collision-free points, i.e. , there is not clearly explained how these points are selected or which kinds of the points is suitable. In the smooth parts, the authors use some tangent circles of the piece-wise line to create a smooth path for UAVs, could you please describe more on how to select the best tangent circle from the viewpoint of mathematics. I still have a question about the 3D path generator, if there are four points (0,0,0), (1,0,0), (1,1,0) and (1,1,1), there are two methods to generate the smooth path, one is using two 2D tangent circles and the other one is using a 3D tangent So from the viewpoint of authors which methods are better? A bibliographic comment: the model presented is very reminiscent of models for space trajectory generate and track, there are some other methods on it by using polynomial method based on discrete points (see e.g. Satellite attitude slew manoeuvres using inverse control Aeronaut. J. (1998); Reconfiguring smart structures using approximate heteroclinic connections, Smart Materials and Structures) It could be very useful if the authors will frame their contribution in this relevant literature.Author Response
We would like to thank the reviewers for their time and efforts on reviewing our submission. It is a pleasure to receive support from experienced professionals who express such accurate opinions. We have modified the manuscript with the aim of solving each of reviewer’s comments and suggestions, emphasizing the changes on the modified manuscript.
Response to Reviewer 2
Point 1: The key parts of the approach in this work is the set of collision-free points, i.e. , there is not clearly explained how these points are selected or which kinds of the points is suitable.
Response 1: Lines (45 – 50) have been modified.
“The set of control points that define the collision-free space is previously calculated using specific path planning methods based on continuous and discrete environment sampling. Some examples of these techniques are: Rapidly-Exploring Random Tree (RPT) [20–23]; Probabilistic Road Maps (PRM) [24–28]; Heuristic planners (Genetic Algorithms-GA) [29,30]; Swarm intelligence [31–34]; Fuzzy logic [35,36]); Voronoi diagrams [37–39]; Artificial Potential [40–43] or Recursive Rewarding Modified Adaptive Cell Decomposition (RR-MACD) [44]. “
Point 2: In the smooth parts, the authors use some tangent circles of the piece-wise line to create a smooth path for UAVs, could you please describe more on how to select the best tangent circle from the viewpoint of mathematics.
Response 2: Lines (197 – 203) have been modified.
“In summary, the selection of ρgoal points which determine radius of the tangent curves to the ρ points, will be obtained by solving a Multiobjective Optimization Problem (MOP). This MOP is stated in such a way that the value of all radius will be maximized simultaneously. Obviously, the optimizer handles these values taking into account they are in conflict (as radius of one of them is increased, consequently, the adjacent radius are reduced). Therefore, MOP solver will try to find a trade-off solution that guarantees the best set of points ρgoal for all tangent curves between control points ρ.”
Point 3: I still have a question about the 3D path generator, if there are four points (0,0,0), (1,0,0), (1,1,0) and (1,1,1), there are two methods to generate the smooth path, one is using two 2D tangent circles and the other one is using a 3D tangent So from the viewpoint of authors which methods are better?
Response 3: Since the proposed study is the generation of smooth 3D trajectories in the space (X,Y,Z) for UAVs, the most convenient method is the direct definition of the problem in 3D. In addition, since the tangent radius is obtained through Multiobjective Optimization, it is not equivalent to minimize the radius with a 2D and 3D approach, so it is necessary to formulate the optimization problem directly in 3D. Obviously, if you consider a fixed altitude, then you might use the generation of a 2D curve (similar to those used in mobile robotics).
Point 4: A bibliographic comment: the model presented is very reminiscent of models for space trajectory generate and track, there are some other methods on it by using polynomial method based on discrete points (see e.g. Satellite attitude slew manoeuvres using inverse control Aeronaut. J. (1998); Reconfiguring smart structures using approximate heteroclinic connections, Smart Materials and Structures) It could be very useful if the authors will frame their contribution in this relevant literature.
Response 4: The comment has been taken into account and added in the section future works - Lines (437–443):
“Connected with the results shown in this paper, several future works arise. For example, integration of dynamic obstacles (UAVs swarms or other aircraft systems) into the flying area. From the optimization point of view, the proposal can be improved taken into account dynamic constraints (i.e, inertia, wind disturbances, torque forces, etc.) into the MOP problem. Similarly, new cost indexes as flight time or/and spent energy could be added to the optimization problem. Finally, implementation of the proposed algorithm under real conditions (UAV in an outdoor environment) and application to different uses (such as satellites trajectories generation [82]) will be investigated.”

Reviewer 3 Report
The paper is too long. The first part presents rather a lecture on differential geometry and optimization theory than scientific raport. In result, some equations are repeated, for example see (12,13) and (27). The path planning for fixed-wing UAVs in the known/structured environment with given control points of the path was fomulated and considered in many papers. It has been also realized for piloted aircraft. Proposed method should be compared with other solutions. The kinematics problem of path finder were only solved in a classic way (geometry+optimization of geometric variables). The dynamics problems (inertia, wind, control torques and forces, etc.) were ommitted what leads to the raw/introductory solutions. It is much more interesting to optimize the dynamic parameters like time or energy. The computer simulations were indicated as an experiment. True lab and field experiments are much more desired.Author Response
We would like to thank the reviewers for their time and efforts on reviewing our submission. It is a pleasure to receive support from experienced professionals who express such accurate opinions. We have modified the manuscript with the aim of solving each of reviewer’s comments and suggestions, emphasizing the changes on the modified manuscript.
Response to Reviewer 3
Point 1: The paper is too long.
Response 1: Figures 3b, 3d and 10c have been removed in order to reduce the extension of the paper.
Point 2: The first part presents rather a lecture on differential geometry and optimization theory than scientific raport. In result, some equations are repeated, for example see (12,13) and (27).
Response 2: (modified after line 324)
Point 3: La planificación del trayecto de los vehículos aéreos no tripulados de ala fija en el entorno conocido/estructurado con puntos de control determinados del trayecto fue fomentada y considerada en muchos documentos. También se ha realizado para aeronaves pilotadas. El método propuesto debe compararse con otras soluciones.
Response 3: The proposed solution has been compared with a classical method based on the generation of trajectories using Bezier curves [77], as indicated in section 5.1 and shown in the results of graphs 8a, 9a, 9b, 9c and 9d.
Point 4: The kinematics problem of path finder were only solved in a classic way (geometry+optimization of geometric variables). The dynamics problems (inertia, wind, control torques and forces, etc.) were ommitted what leads to the raw/introductory solutions. It is much more interesting to optimize the dynamic parameters like time or energy. The computer simulations were indicated as an experiment. True lab and field experiments are much more desired.
Response 4: The comment has been taken into account and added in the section future works - Lines (437–443):
“Connected with the results shown in this paper, several future works arise. For example, integration of dynamic obstacles (UAVs swarms or other aircraft systems) into the flying area. From the optimization point of view, the proposal can be improved taken into account dynamic constraints (i.e, inertia, wind disturbances, torque forces, etc.) into the MOP problem. Similarly, new cost indexes as flight time or/and spent energy could be added to the optimization problem. Finally, implementation of the proposed algorithm under real conditions (UAV in an outdoor environment) and application to different uses (such as satellites trajectories generation [82]) will be investigated.”
